# Modeling and Phenotyping Acute and Chronic Type 2 Diabetes Mellitus In Vitro in Rodent Heart and Skeletal Muscle Cells

**DOI:** 10.3390/cells12242786

**Published:** 2023-12-07

**Authors:** Elena L. Kopp, Daniel N. Deussen, Raphael Cuomo, Reinhard Lorenz, David M. Roth, Sushil K. Mahata, Hemal H. Patel

**Affiliations:** 1Department of Anesthesiology, University of California San Diego, San Diego, CA 92161, USA; 2Faculty of Medicine, University of Munich (LMU Munich), 80539 Munich, Germany; 3Institute for Cardiovascular Prevention (IPEK), LMU Munich, 80539 Munich, Germany; 4VA San Diego Healthcare System, San Diego, CA 92161, USA; 5Department of Medicine, University of California, San Diego, CA 92093, USA

**Keywords:** insulin resistance, type 2 diabetes, cell model, mitochondrial dysfunction

## Abstract

Type 2 diabetes (T2D) has a complex pathophysiology which makes modeling the disease difficult. We aimed to develop a novel model for simulating T2D in vitro, including hyperglycemia, hyperlipidemia, and variably elevated insulin levels targeting muscle cells. We investigated insulin resistance (IR), cellular respiration, mitochondrial morphometry, and the associated function in different T2D-mimicking conditions in rodent skeletal (C2C12) and cardiac (H9C2) myotubes. The physiological controls included 5 mM of glucose with 20 mM of mannitol as osmotic controls. To mimic hyperglycemia, cells were exposed to 25 mM of glucose. Further treatments included insulin, palmitate, or both. After short-term (24 h) or long-term (96 h) exposure, we performed radioactive glucose uptake and mitochondrial function assays. The mitochondrial size and relative frequencies were assessed with morphometric analyses using electron micrographs. C2C12 and H9C2 cells that were treated short- or long-term with insulin and/or palmitate and HG showed IR. C2C12 myotubes exposed to T2D-mimicking conditions showed significantly decreased ATP-linked respiration and spare respiratory capacity and less cytoplasmic area occupied by mitochondria, implying mitochondrial dysfunction. In contrast, the H9C2 myotubes showed elevated ATP-linked and maximal respiration and increased cytoplasmic area occupied by mitochondria, indicating a better adaptation to stress and compensatory lipid oxidation in a T2D environment. Both cell lines displayed elevated fractions of swollen/vacuolated mitochondria after T2D-mimicking treatments. Our stable and reproducible in vitro model of T2D rapidly induced IR, changes in the ATP-linked respiration, shifts in energetic phenotypes, and mitochondrial morphology, which are comparable to the muscles of patients suffering from T2D. Thus, our model should allow for the study of disease mechanisms and potential new targets and allow for the screening of candidate therapeutic compounds.

## 1. Introduction

Diabetes mellitus (DM) represents a large global disease burden with an ominous prognosis [1]. It is predicted that more than 550 million people worldwide will be diabetic by 2030, compared to 108 million in 1980 [2]. Diabetes is among the diseases with the most rapidly increasing global prevalences, and improving therapies and prevention methods to reduce diabetes-related complications and premature mortality is of utmost importance.

DM is characterized by a relative or absolute lack of insulin, resulting in elevated blood glucose levels. Type 2 diabetes (T2D) accounts for more than 90% of DM cases and is primarily caused by an inappropriate cellular response to insulin. In response to insulin resistance (IR) or intolerance, pancreatic beta cells undergo hypertrophy and release more insulin, consequently leading to pancreatic exhaustion and failure in later stages of the disease [3]. The natural history of T2D is characterized by obesity in >80% of cases, which is further exacerbated by a Western diet and physical inactivity, leading to IR, hyperglycemia, hyperlipidemia, and hyperinsulinemia [4]. Chronic hyperglycemia can lead to microvascular (e.g., nephropathy, retinopathy, and polyneuropathy) and macrovascular (e.g., stroke and cardiovascular disease) complications [5]. In T2D patients, cardiovascular disease is a major cause of morbidity and mortality, accounting for 68% of all diabetes-related deaths [6]. 

Given the physiological complexity of the disease, in vitro models that could provide a path to the discovery and testing of novel therapeutics have been limited. By modeling this complex disease in vitro, we want to increase and accelerate the discovery of therapeutic targets, improve disease screening, and enable improved preclinical drug testing. In vitro cell models can provide an economically and ethically acceptable research tool and enable the targeting of specific processes linked to a single cell type of interest without uncontrolled influences of the whole organism. In vitro cell models can be performed with high throughput and reproducibility and can be genetically modified by transfection to investigate specific genes of interest [7]. Thus, cell-based models, if developed properly, could serve as tools to accelerate therapeutic discovery.

In general, in vitro models of diabetes are derived from the tissues that are mainly involved in the disease pathophysiology, such as the muscle, pancreas, adipose tissue, and liver [8]. Skeletal muscle is the primary recipient of postprandial glucose, and T2D patients often show significantly impaired blood glucose clearance [9]. Also, the higher flux of FFAs in patients with T2D leads to an increased FA uptake into myocytes [10]. To assess the effect of T2D on skeletal muscle, we chose to use C2C12 cells, a mouse myoblast cell line, which is well established and has been widely used in diabetes research [11]. In the setting of IR, the myocardium’s ability to utilize glucose as an energy source is reduced [12], whereas under physiological conditions, energy utilization from FA and carbohydrates is based on metabolic demand and availability [13]. Even though FFAs are the major energy source for heart muscle, an excess of plasma FFAs may predispose the heart to ischemic damage and high oxidative stress [14]. These metabolic changes and the accumulation of lipids in the heart play critical roles in the development of diabetes-related cardiac complications, such as diabetic cardiomyopathy [15,16]. To further understand these mechanisms in the heart muscle, we mimicked diabetic conditions in H9C2 cells, which are myocytes from embryonic rat ventricular tissue [17]. H9C2 myocytes are widely applied in studies addressing cardiac hypertrophy, metabolism, ischemic stress, and IR/diabetes [18,19].

To cause IR as it occurs in T2D, most cell models with adipocytes and myocytes use chronic insulin exposure or fatty acid treatments (mainly palmitate) [20] (Table 1). The most commonly used immortalized muscle cell lines for in vitro diabetes models are L6 and C2C12 myoblasts [21].

To simulate the diabetic milieu in vitro, not only do the cells themselves play critical roles, but also the conditions in which they are cultured. The main reason for metabolic syndrome and T2D is chronic overnutrition consisting mainly of fatty acids and carbohydrates in the developed world [32]. In vitro studies showed that once a cell becomes overwhelmed by the excess supply of FFAs, the accumulation of lipids such as diacylglyceride (DAG) and ceramide may contribute to mitochondrial dysfunction, the generation of reactive oxygen species (ROS), low-grade inflammation, and finally, IR and apoptosis [33]. Correspondingly, symptoms of T2D are caused by both hyperglycemia and hyperlipidemia, whereas hyperglycemia alone would more likely reflect uncontrolled T1D [34]. In cell culture research, it is a common practice to use culture media with supraphysiological glucose concentrations to promote and accelerate cellular growth. Previous studies addressing IR and diabetes in H9C2 and C2C12 muscle cells worked with 22–25 mM of glucose as a control and 33–40 mM of glucose to mimic hyperglycemia, which is eight times more than physiological levels [28,29]. In several studies investigating DM, the glucose concentrations of the cell culture media are not specified, leaving the reader uninformed (Table 1) [26,27,31]. The American Diabetes Association and the WHO define normoglycemia as a fasting plasma glucose between 3.9 mmol/L and 5.6 mmol/L [35,36]. Diabetes is defined when the fasting plasma glucose reaches 7.0 mmol/L or higher [37]. It is recommended to first grow and differentiate cells in physiological glucose levels of 5 mM [25]. The simulation of hyperlipidemia requires a treatment with fatty acids (FAs). Chavez and Summers showed that the saturated FA palmitate (the most dominant fat in the Western fast food diet) causes IR, whereas unsaturated FAs such as oleate even reverse palmitate-induced IR in skeletal muscle [20]. This suggests that the widely applied FA palmitate can be used for T2D-related hyperlipidemia conditions. High concentrations of palmitate (200–750 µM), which have previously been used in studies investigating IR, can cause apoptosis and high myotube loss in skeletal and cardiac muscle cells (see Appendix A) [26,38,39]. We therefore worked with a maximal concentration of 150 µM for C2C12 and 75 µM for H9C2, which still leads to IR but is less toxic and close to the physiological range (300–410 µM in humans [40] and 100–400 µM in rats [41]). In conclusion, exposure to high glucose, palmitate, and insulin together may be closer to the conditions observed in T2D patients compared to the previously described cell-based approaches (Table 1). To achieve more similarity to human muscle tissue, we worked with differentiated myotubes, whilst many research groups work with undifferentiated myoblasts [25,42,43]. The insulin-dependent glucose transporter 4 (GLUT4) is similarly expressed in C2C12 myotubes compared to human skeletal muscle cells [11]. Undifferentiated H9C2 myoblasts express little cardiac-specific markers and can still differentiate into skeletal muscle instead of cardiac muscle. In 2017, Patten et al. stated that the combination of serum reduction with retinoic acid (RA) supplementation in the differentiation process increased the expression of cardiac-specific markers [44]. Lopashuk et al. identified that only differentiated H9C2 cells switch from glycolysis to oxidative phosphorylation, which is characteristic of heart tissue [45]. Also, insulin-signaling molecules such as insulin receptor substrate-1 and GLUT4 are significantly more expressed by differentiated H9C2 myotubes [46]. In conclusion, differentiation into myotubes to explore heart- and skeletal muscle-specific processes related to T2D is necessary. Various human studies suggested the existence of mitochondrial dysfunction in the muscle cells of obese and insulin-resistant patients [47,48]. A comprehensive understanding of the mechanisms that relate mitochondrial function and insulin signaling is still lacking. Identifying the factors and mechanisms responsible for the changes in mitochondrial energetics could make mitochondria a potential target for treating diabetes. Therefore, we examined mitochondrial function in vitro in rodent skeletal and heart muscle utilizing the Agilent Seahorse XF Analyzer and utilized transmission electron microscopy (TEM) to examine mitochondrial morphometry.

Taken together, this study aimed to develop a novel model for simulating T2D in vitro in skeletal and cardiac muscle cells that reflects the cellular response typified by T2D patients.

## 2. Materials and Methods

### 2.1. Cell Culture 

C2C12 (CRL-1772™) and H9C2 (CRL-1446™) cell lines were purchased from American Type Culture Collection (ATCC, Manassas, VA, USA). C2C12 and H9C2 cells were cultured in Dulbecco’s Modified Eagle’s Medium (DMEM) (GIBCO/BRL, Waltham, MA USA, 11885092) containing 5 mM glucose, supplemented with 10% fetal bovine serum (FBS, Thermo Fisher Scientific, Waltham, MA, USA, 16000044) and 1% penstrep (Penicillin/Streptomycin, GIBCO, Waltham, MA USA, 15140122). For differentiation of C2C12, DMEM (5 mM glucose) was supplemented with 2% horse serum (HS) (Thermo Fisher, Waltham, MA, USA, 26050088), and 1% penstrep was refreshed daily. For differentiation of H9C2 cells, DMEM (5 mM glucose) was supplemented with 1% FBS, 1% penstrep, and 10nM all-trans-retinoic acid (RA) (Sigma Aldrich, St. Lois, MO, USA, R2625-500MG). RA was diluted in DMSO and stored at −20 °C in the dark. Medium change was performed daily in the dark. Cell treatments for assays were started 5–8 days after differentiation. C2C12 and H9C2 cell cultures were used in passages 2–6 in all experiments.

### 2.2. Treatments 

DMEM containing 25 mM glucose (GIBCO/BRL, Waltham, MA, USA, 11995073) was used as a hyperglycemic condition (HG). This hyperglycemic condition was also used in combination with insulin (HG-I) (1 nM, insulin glargine, Lantus, Sanofi, Paris, France) or palmitate (HG-P) (150 µM for C2C12, 75 µM for H9C2). A further hyperglycemic condition was combined with both insulin and palmitate (HG-PI). Groups treated with low glucose (LG, 5 mM) and the same combinations with palmitate and/or 1 nM insulin served as physiological controls (LG-I, LG-P, and LG-PI). To exclude a possible hyperosmolar effect, cells treated with 20 mM mannitol plus 5 mM glucose served as control cultures (high mannitol, HM). The same treatment combinations were used (HM-I, HM-P, and HM-PI). The cells were exposed to different conditions, either short-term (24 h) or long-term (96 h). Before all experiments, the culture medium was changed to serum-free medium for 4 h. For long-term treatments, the concentration of palmitate was escalated every 24 h to prevent myotube loss (Table 2). 

Before adding sodium palmitate to cell differentiation medium, it was conjugated to fatty-acid-free bovine serum albumin (BSA). Transportation into cells was then enabled, and cytotoxicity was lowered. Briefly, ultra-fatty-acid-free BSA (Sigma Aldrich, St. Lois, MO, USA, A8806) was dissolved in the respective differentiation medium for each cell line, heating the solution to approximately 37 °C. Sodium palmitate (Sigma Aldrich, St. Lois, MO, USA, P9767) was dissolved in a 150 mM NaCl solution, stirring at 70 °C. The palmitate solution was added to the BSA solution while stirring at 37 °C. After stirring for 1 h, pH was adjusted to 7.4 with NaOH. The conjugated palmitate-BSA solution was aliquoted in glass vials and frozen at −20 °C. Stock solutions were made at 75 mM.

### 2.3. ^3^H-2-Deoxy-Glucose-Uptake 

^3^H-2-deoxy-glucose-uptake assays were performed using published methods [49]. Briefly, culture media were discarded 4 h prior to the assay and changed to serum-free DMEM with 0.25% fatty-acid-free BSA. Serum starvation was held in respective treatment media (serum-free DMEM with either high or low glucose (5 mM vs. 25 mM) and supplemented with palmitate or 1 nM insulin, respectively). After starvation, cultures were washed twice with Hepes-fortified Krebs–Ringer Bicarbonate buffer (HKRB), containing 10 mM Hepes, pH 7.4, NaH_2_PO_4_ (0.83 mM), Na_2_HPO_4_ (1.27 mM), NaHCO_3_ (15 mM), NaCl (120 mM), KCl (4.8 mM), calcium (1 mM), magnesium (1 mM), pH 7.35, and 0.25% fatty-acid-free BSA. Cultures were incubated in HKRB for 60 min at 37 °C. Half of the cultures were pre-assigned for acute insulin treatment. These cultures were treated with 100 nM insulin from a 100x stock and incubated for 20 min at 37 °C. After this incubation period, ^3^H-2-deoxy-glucose (^3^H-2DOG, 2-[1,2-3H(N)]-, 250 µCi (9.25 MBq), Perkin Elmer, Waltham, MA, USA, NET549A250UC) was added to each well for further 10 min incubation. ^3^H-2DOG-treatment was terminated by quick aspiration, followed by cold wash (2×) with ice-cold PBS. Next, we added 1 N NaOH and swirled the plates slowly for 30 min to dissolve the cells. Aliquots of 20 µL were taken out of each well for protein measurements. The whole lysate of each well was transferred into a scintillation vial. After neutralizing NaOH with 1 N HCl, scintillation cocktail (Ecoscint original, National Diagnostics, Brea, CA, USA, LS-271) was added and radioactivity was measured. Results were normalized with protein concentrations and presented as counts per minute (cpm) per mg protein. Alternately, the results were presented as cpm per million cells.

### 2.4. Mitochondrial Function Measurements

Mitochondrial function measurements were performed with the Agilent Seahorse XFe 96 Extracellular Flux Analyzer (Agilent Technologies, La Jolla, CA, USA). The analyzer measures real-time extracellular acidification rate (ECAR) and oxygen consumption rate (OCR) of cells. C2C12 and H9C2 cells were seeded and differentiated in a density of ~1.2 × 10^4^ cells per well, and the Seahorse XF Mito Stress Test assay was carried out as previously described [50]. During the assay, inhibitors of mitochondrial function were injected sequentially to distinguish the following parameters: Basal respiration: Oxygen consumption used to meet cellular ATP demand under baseline conditions. It can be set by the rate of ATP utilization and substrate availability and oxidation. ATP-linked respiration: Upon injection of the ATP synthase inhibitor, oligomycin, a decrease in OCR represents the part of basal respiration that accounts for ATP-linked respiration, meeting the energetic needs of the cell. It is largely set by the ATP demand of the cell and substrate oxidation, which can be decreased if there is mitochondrial dysfunction. Maximal respiration: By adding the uncoupler carbonyl cyanide p-(trifluoromethoxy), phenylhydrazone (FCCP), the maximal OCR can be attained. FCCP mimics an “energy demand” and stimulates the respiratory chain to operate at maximum capacity, and substrates are rapidly oxidized. It is also set by substrate supply and oxidation, including the functional substrate transport across the plasma and mitochondrial membranes. Changes may reflect membrane defects, altered mitochondrial biogenesis and function, and decreased oxidation abilities. Spare respiratory capacity: This value can be calculated with basal and maximal respiration values. It indicates the cell’s capability to respond to an energetic demand and can be an indicator of cell fitness or flexibility. A decreased capacity can be linked to mitochondrial dysfunction but can also reflect an increased ATP demand/increased basal respiration (i.e., highly proliferative cells). Increases may reflect high substrate provision or enhanced oxidative capacity [51]. We plotted OCR versus ECAR values on an energy map to provide a qualitative measurement of the relative utilization of oxidative (OCR) versus glycolytic (ECAR) pathways for energy production.

### 2.5. Electron Microscopy

C2C12 and H9C2 myotubes maintained short-term (24 h) and long-term (96 h) in different T2D-mimicking and control conditions (see: treatments) were fixed and embedded for electron microscopy (EM) as previously described [52]. Grids were viewed using a Jeol JEM1400-plus Transmission Electron Microscope (JEOL Ltd., Akishima City, Janpan) and photographed with a Gatan OneView digital camera (Gatan, Pleasanton, CA, USA) with 4k × 4k resolution.

### 2.6. Morphometric Analyses of Mitochondria 

Using the free-hand tool of NIH ImageJ (Version 1.54f, Rasband, W.S., Bethesda, MD, USA) we manually traced around the mitochondria to determine mitochondrial area and length, as described previously [53]. Cytoplasmic area (excluding the area occupied by the nucleus) was also calculated using the ImageJ software. We then calculated mitochondrial density (area occupied by mitochondria in total area of cytoplasm). Mitochondrial numbers were calculated by placing the EM micrographs onto a grid with 10 horizontal × 10 vertical square grids followed by dividing the mitochondrial numbers with the area. The data were expressed as mitochondrial number/10 µm^2^ area.

### 2.7. Statistical Analysis

All data analyses were performed using GraphPad Prism 9.5.1 (GraphPad Software, La Jolla, CA, USA). For glucose uptake assays, the results are expressed as mean ± SD of 6 biological replicates for all LG and HG groups and 3 replicates for all HM groups. For Mito Stress Tests, all data shown are the means ± SD of 3–4 replicate plates with 8–15 wells per treatment group per plate. Relative frequencies of mitochondria area and length were analyzed using Kolmogorov–Smirnov test; mitochondrial number and density were analyzed with one-way ANOVA of 6–14 cells. For statistical analysis, a *p* value of <0.05 was considered significant. We used analysis of variance (one-way or two-way ANOVA), provided that the assumptions were fulfilled. For Agilent Seahorse measurements, two-way ANOVA was used with row factor for treatment groups (Rf) and column factor for biological replicates/assays (Cf). F- and *p* values are provided in the legend of the figure panels. Significance of secondary pairwise comparisons among different treatments by Tukey’s test is indicated by asterisks over brackets in the figure panels (*: *p* < 0.05; **: *p* < 0.01; ***: *p* < 0.001).

## 3. Results

### 3.1. ^3^H-2-Deoxy-Glucose-Uptake

C2C12 myotubes: We found that the treatment of C2C12 myotubes with T2D-mimicking conditions including insulin and palmitate treatments resulted in IR compared to the controls. The control cultures preincubated with LG for 24 h showed a significant increase in the glucose uptake upon stimulation with 100 nM of test insulin (Figure 1A) (LG +/− insulin). The short-term (24 h) pretreatments with 150 µM of palmitate either in the presence or absence of 1 nM of insulin impeded the stimulatory effect of the insulin, indicating that palmitate caused IR in consistence with the existing literature [54]. Similar effects were seen after 96 h (Figure 1K). After short- and long-term treatments, the basal glucose uptake was increased in the LG-I treated groups compared to the LG controls (Figure 1A,K). The LG-P- and LG-PI-treated myotubes displayed decreased glucose uptake rates compared to the LG and LG-I treatments. The cultures that were treated short- or long-term in an HG condition did not increase their glucose uptakes upon insulin stimulus (Figure 1C,M) but showed elevated basal rates compared to the LG. We did not observe a significant increase in the insulin-dependent vs. basal glucose uptakes in any of the T2D-mimicking treatment groups (HG, HG-I, HG-P, and HG-PI). The myotubes that were treated with HG-P or HG-PI showed significantly lower basal and insulin-dependent glucose uptake rates compared to the HG and HG-I groups (Figure 1C,M). The myotubes that were treated with HM or HM-I for 24 h showed a significant decrease in the basal glucose uptake compared to the LG or HG, and there was no significant up-regulation in the glucose uptake upon insulin stimulation. Interestingly, the C2C12 myotubes treated with HM and palmitate (HM-P and HM-PI) showed a marked increase in both the basal and insulin-dependent glucose uptakes (Figure 1G) compared to all other treatment groups. Looking at the half-violin plots with the combined data of the 24 h and 96 h treatments, the treatments including mannitol show the widest spread in values due to the stimulatory effect on the glucose uptake of palmitate in combination with HM (Figure 1E), whereas in the cells that were exposed to HG and palmitate, an inhibitory effect of palmitate is apparent (Figure 1I).

H9C2 cardiac myotubes: IR occurred in all T2D-mimicking treatment groups. The LG- or LG-I treated control cultures showed an increase in the glucose uptake upon an acute insulin stimulus (100 nM). After 24 h of treatment, the increase was not significant, whereas after 96h of treatment, we observed a significant rise in the insulin-dependent glucose uptake compared to the basal uptake (Figure 1B,L). The basal glucose uptake rates in both the LG and LG-I treatment groups were similar, but the insulin-dependent rates were significantly higher in the LG-I group compared to the basal rates in the LG group (Figure 1B). The H9C2 myotubes that were treated short- or long-term with LG and palmitate (LG-P and LG-PI) showed significantly decreased glucose uptake rates compared to LG and LG-I. We observed similar effects on the glucose uptake after short- or long-term treatments with HG-P or HG-PI (Figure 1D,N), which indicates a massive suppression of basal glucose influx and insulin insensitivity by palmitate [31]. The groups that were treated short- or long-term with HG or HG-I were at least partially insulin-resistant and showed significantly higher glucose uptakes compared to the groups treated with high glucose and palmitate (Figure 1D,N). In contrast, the osmotic control cultures treated with HM-P or HM-PI showed a remarkable increase in the basal glucose uptake compared to the HM or HM-I treatment, but the insulin response was suppressed (Figure 1H) even more marked than in the C2C12 myotubes (Figure 1G). In summary, the HM-treated groups showed markedly lower glucose uptake rates compared to the LG groups (LG basal rate mean = 207.33; HM basal rate mean = 45.95). The cultures treated with HM or HM-I showed a non-significant trend towards an increase in the glucose uptake upon an acute insulin stimulus (Figure 1H). Looking at the half-violin plots with the combined data of 24 h and 96 h treatments, the cells that were exposed to mannitol show the lowest glucose uptake rates of all treatments (Figure 1F). Overall, palmitate exerted a marked inhibitory effect on both basal- and insulin-dependent glucose uptake, particularly in combination with the HG treatment (Figure 1J,P).

### 3.2. Mitochondrial Function Measurements Using the Agilent Seahorse XFe 96 Extracellular Flux Analyzer

In all Mito Stress Test assays with C2C12 and H9C2 myotubes, baseline relative oxygen consumption rate percentage did not differ statistically (Figure 2D,H,L,P).

#### 3.2.1. C2C12 Myotubes

Treatments for 24 h: C2C12 myotubes that were treated for 24 h with non-physiological glucose levels (HG, HG-I, HG-P, and HG-PI) showed a significant decrease in ATP-linked respiration compared to the LG and LG-I controls (Figure 2A). The osmotic control cultures (HM, HM-I) did not show differences to the LG control groups in maximal respiration (Figure 2B), but they showed a significant decrease in ATP-linked respiration (Figure 2A). In all groups where the treatments included insulin, the maximal respiration rates were higher compared to the treatments without insulin (Figure 2B). The spare respiratory capacity rates were significantly increased in the HG-I and HM-I groups compared to the LG control (Figure 2C). In the short-term assays with the C2C12 myotubes, we observed a shift towards glycolysis in all of the HG-treated groups. The HM-treated groups showed a metabolic shift towards glycolysis. The LG groups showed a mainly aerobic metabolism, and the LG-I groups were in between an aerobic and high energetic phenotype (Figure 2Q). 

Treatments for 96 h: C2C12 myotubes that were treated for 96 h with high glucose (HG and HG-I) or HM-I showed a significant decrease in maximal respiration compared to the LG control (Figure 2F). The ATP-linked respiration of the myotubes treated with HG or HM was significantly decreased (Figure 2E). The spare respiratory capacity rates were significantly decreased in the HG- and HG-I-treated groups compared to the LG. The HM- and HM-I-treated groups displayed a lower spare respiratory capacity compared to LG-I (Figure 2G). In the XF energy map, the HG- and HG-I-treated groups showed a more glycolytic phenotype, while the LG and LG-I control groups were in between an aerobic (oxidative phosphorylation) and a high energetic phenotype. The HM and HM-I groups were less energetic compared to LG and less glycolytic compared to HG (Figure 2R). All respiration rates were lower after 96 h of treatment compared to the 24 h treatment, with maximal respiration rates being remarkably lower in the T2D-mimicking conditions after 96 h compared to the 24 h treatment. After the 96 h treatment with HG-P and HG-PI, we did not observe a significant difference to the LG controls in one Mito Stress Test and therefore excluded these treatment groups in the following repetitions.

#### 3.2.2. H9C2 Myotubes

Treatments for 24 h: H9C2 myotubes that were treated for 24 h with high glucose in combination with insulin or palmitate (HG-I, HG-P, and HG-PI) showed a significant increase in the ATP-linked respiration and maximal respiration compared to the LG and LG-I controls, with the HG-PI groups displaying the significantly highest ATP-linked respiration and respiration rates (Figure 2I,J). The spare respiratory capacity was significantly increased in the treatment groups, including palmitate (HG-P and HG-PI) (Figure 2K). Plotting the OCR vs. ECAR showed a trend towards more energetic phenotypes in the T2D-mimicking conditions compared to the LG controls (Figure 2S). All treatment groups utilized both glycolysis and oxidative phosphorylation to generate energy. The high mannitol control cultures did not show significant differences in all of the respiration rates compared to the LG control groups in one Mito Stress Test assay and were therefore excluded in the following assay repetitions. 

Treatments for 96 h: H9C2 myotubes that were treated for 96 h with high glucose in combination with insulin or palmitate (HG-I, HG-P, and HG-PI) showed a significant increase in the ATP linked respiration and maximal respiration compared to the LG and LGI controls (Figure 2M,N). After the 96h treatment, no significant changes in the spare respiratory capacity were detectable (Figure 2O). Plotting the OCR vs. ECAR respiration rates after the 96 h treatment revealed the LG control cultures to be more quiescent compared to the 24 h treatment and compared to the T2D-mimicking conditions. The HG-, HG-I-, HG-P-, and HG-PI-treated groups showed a phenotype in between glycolysis and oxidative phosphorylation, with HG-PI being the most energetic (Figure 2T).

### 3.3. Morphometric Analyses of Mitochondrial Number, Density, and Relative Frequency of Mitochondria with Differing Area and Length

#### 3.3.1. Electron Microscopy for Mitochondrial Appearance

The abnormalities caused by the T2D conditions in mitochondrial function raised the question of accompanying structural abnormalities. For a mitochondrial morphology analysis, the number of normal versus swollen/vacuolated mitochondria was compared after the 24 h treatment of C2C12 and H9C2 myotubes. Normal mitochondria were defined as indicated in Figure 3A, and swollen/vacuolated mitochondria were defined as indicated in Figure 3B,C. The HM-, HG-P-, and HG-PI-treated C2C12 myotubes showed a significantly higher fraction of swollen/vacuolated mitochondria compared to the LG control (Figure 3K). The fraction of swollen/vacuolated mitochondria in the HG groups compared to the LG groups was not significantly different (Figure 3K). Compared to LG, the percentage of swollen/vacuolated mitochondria in the H9C2 myotubes was higher in the HM, HG, and HG-PI groups (Figure 3S). Swollen mitochondria are marked as “sm” in Figure 3 in the C2C12 HM, HG-P, and HG-PI treatments (Figure 3E,G,H) and the H9C2 HM, HG, and HG-PI treatments (Figure 3M,N,Q).

#### 3.3.2. Mitochondrial Number and Density

The morphometric analyses did not reveal significant changes in the mitochondrial number per 10 µm^2^ area of cytoplasm after exposure to T2D conditions in both the C2C12 (Figure 3I) and H9C2 (Figure 3O) cells.

Opposite changes in the mitochondrial density between the C2C12 and H9C2 cells: The morphometric analyses of the mitochondrial density (area occupied by mitochondria in total area of cytoplasm) revealed a significant decrease in the mitochondrial density when the C2C12 cells were exposed to HG, HGP, or HGPI compared to LG (Figure 3J). In contrast, the H9C2 cells showed a significant increase in the mitochondrial density when exposed to HGP or HGPI compared to LG and HG (Figure 3R). These findings are consistent with the increase in the ATP-linked respiration, as described above.

#### 3.3.3. The Combination of Palmitate with High Glucose Changed Relative Frequency of Mitochondria with Respect to Area and Length

The morphometric analyses revealed that HG alone did not alter the relative frequency of mitochondria with respect to the area and length in both the C2C12 and H9C2 cells compared to LG (Figure 4A,J). The combination of palmitate with high glucose caused marked changes in the relative frequency of mitochondria with respect to the area and length in both the C2C12 and H9C2 cells compared to the LG and HM controls (Figure 4B,F,K,O).

#### 3.3.4. Insulin Failed to Alter Palmitate-Induced Changes in Mitochondrial Health

We did not find significant changes in the mitochondrial area and length between the HGP and HGPI groups in both the C2C12 and H9C2 cells (Figure 4H,Q), indicating that palmitate in combination with HG caused profound IR in muscle cells.

## 4. Discussion

We observed that complex T2D in in vitro modeling has significant effects on the functionality of rodent skeletal and heart muscle cells, leading to severe IR, changes in ATP-linked respiration, shifts in energetic phenotypes, and mitochondrial morphology changes, which are all consistent with what has been observed in the muscles of patients suffering from T2D [55]. Pre-existing in vitro models of T2D often disregard many critical and complex aspects of the disease (Table 1). The current study shows that by following some essential steps lacking in the literature, a representative in vitro model of human T2D-related changes in muscle cells can be created.

Basal glucose uptake rates were increased in both cell lines after HG treatment compared to LG, especially after long-term exposure. The gradient-driven glucose uptake might have facilitated this via GLUT1 instead of insulin-dependent GLUT4. McMillin et al. showed that mGLUT4 knockout mice still showed increased basal glucose uptake after chronic exposure to HG. They also found that in mouse skeletal muscle cells, GLUT1, 3, 6, or 10 almost exclusively mediated glucose uptake after chronic glucose overload [56]. Gosmanov et al. observed that, compared to 5 mM of glucose exposure, the HG treatment (30 mM, up to 48 h) of aortic endothelial cells increased the GLUT1 expression and GLUT4-dependent glucose uptake, but did not change the baseline glucose uptake rates. Heilig et al. saw a 134% increase in the GLUT1 mitochondrial RNA as well as a 50% increase in the deoxy-glucose uptake in rat mesangial cells that were exposed to 20 mM of glucose for 3 days when compared to cells that adapted to physiological glucose levels (8 mM). 

We examined the mitochondrial function and metabolic phenotypes in rodent skeletal and heart muscle. Skeletal myotubes from insulin-sensitive subjects with T2D family history have decreased ATP contents, which is consistent with previous studies showing decreased ATP and impaired mitochondrial activity in myotubes from lean offspring of T2D patients [57,58,59]. Mailloux et al. also observed that C2C12 myotubes exposed to 24 mM of glucose for 24 h showed a glycolytic phenotype and more ROS production than a low glucose control with an oxidative phenotype; however, the ATP-linked respiration was not altered, indicating that those myotubes were still metabolically flexible and achieved the ATP demand via glycolysis [60]. Elkalaf et al. followed a similar approach and saw differences in the maximal respiration in the hyperglycemic C2C12 myotube cultures and a phenotype switch. In both studies, the cells were differentiated for up to 7 days in media containing 5 mM or 25 mM of glucose before any experiment [61]. Differentiating cells in high glucose might have caused an adaptation of the cells, leading to restored cellular flexibility. In the current study, cells were differentiated in low glucose levels and only exposed to high glucose for 24–96 h when already differentiated, which more accurately represents the changes of T2D in vivo. Future experiments should include a longer exposure to high glucose to investigate if, following differentiation, an adaptation to supraphysiological glucose and correspondingly non-altered ATP levels is possible. 

Strongly increased respiration rates might be associated with elevated cellular stress since high glucose and palmitate levels lead to apoptosis in cardiomyocytes [62,63,64]. On the other hand, it can also reflect a highly energetic phenotype with increased oxidation abilities [51]. The increase in ATP-linked respiration might be associated with an increase in FA oxidation and decreased glucose utilization, which is assumed to be increased in the T2D heart in humans [65,66]. FA uptake into the heart is mainly driven by the availability in the blood stream [67]. With an oversupply in fatty acids, not only is FA oxidation increased, but also detrimental lipid metabolites (i.e., ceramides) [63,68]. The reliance of the heart on FA oxidation to produce ATP might lead to oxidative stress and ischemic damage [14]. Therefore, losing the ability to switch to glycolysis combined with an increase in ROS due to the increased FA oxidation and increasing lipotoxicity contribute to both decreased ATP production and cardiac inefficiency. 

In patients with T2D, the mitochondrial content is reduced, the size and fusion are impaired, and endoplasmic reticulum stress occurs in different cell types [69]. Increased fission and impaired fusion were observed in human renal glomerular endothelial cells treated with high glucose (30 mM for 72 h) [70]. A reduced number and fragmented mitochondria were found in skeletal muscle from T2D and obese subjects, as well as decreased electron transport chain activity [47]. In our study, we observed a significant decrease in the mitochondrial density in diabetic C2C12 cells, which is consistent with decreased ATP-linked respiration. We did not observe any significant changes in the mitochondrial number per 10 µm^2^ area of cytoplasm after 24 h of exposure to T2D-mimicking conditions. Thus, it is likely that the reduction in the mitochondrial number that was reported in patients suffering from T2D is a chronic condition that will only occur in vitro after longer treatments.

The H9C2 myotubes treated with T2D-mimicking conditions including palmitate (HGP and HGPI) showed a significant increase in mitochondrial density compared to the control, which can be a sign of increased fusion, allowing for an enhanced transport of metabolites and enzymes. This observation is consistent with our Mito Stress Tests, where these cells showed increased ATP-linked respiration, confirming a better adaptability to an in vitro T2D environment of heart muscle cells.

Both the C2C12 and H9C2 myotubes treated with HM and T2D-mimicking conditions showed a higher fraction of swollen/vacuolated mitochondria compared to the LG controls. Mitochondrial swelling can be caused by osmotic changes in cell culture media and FA treatments but is also a known sign of apoptosis and necrosis [71]. The question remains whether the decreased ATP-linked respiration we observed in the skeletal muscle cells in the T2D environment led to imminent apoptosis and then mitochondrial swelling, or if the T2D mimicking treatments directly caused mitochondrial swelling, subsequently leading to mitochondrial dysfunction and a decreased ATP turnover.

There are several limitations to consider in our studies. We chose to use the saturated FA palmitate for our investigations, as it was previously used for studies addressing IR. Even when the Western diet is dominated by saturated FAs, using only one saturated FA is not physiologically accurate since in vivo circulating FFAs are a mixture of various saturated and unsaturated FAs. For future experiments, a mixture of the most common FAs in human plasma could be used (i.e., oleic, palmitic, and stearic acids) [72]. 

High mannitol treatments served as the osmotic controls for the high glucose treatments in our study, as it has been widely applied in the literature. Yet, we observed some hitherto undescribed effects, especially after 96 h of high mannitol treatments, such as decreased ATP-linked respiration, a shift to glycolysis but decreased basal glucose uptake, and mitochondrial swelling. The mechanistic background of the effects that we observed must still be explored. Still, it can be stated that the impact of hyperosmolarity and hyperlipidemia in a hyperglycemic T2D environment appears more complex than expected.

## 5. Conclusions

We used two well-established skeletal and cardiac myoblast cell lines for this model and conducted differentiation with close “normoglycemic and normoinsulinemic” conditions, thus avoiding diabetogenic preconditioning before use in experiments. In the experiments, we simulated, in addition to hyperglycemia and hyperinsulinemia, other aspects of T2D, i.e., hyperlipemia and hyperosmolarity. Despite using lower concentrations than previous studies, closer to the range seen in patients with T2D, we demonstrated a very rapid and progressive development of insulin resistance and derangements in the mitochondrial metabolism and morphology, with specific differences between skeletal and cardiac myotubes that parallel the findings in biopsies from T2D patients. The model could potentially be extended to other tissues impacted by diabetes. Our model should therefore help to spare animal experiments and studies requiring human biopsies and has the potential to be applied to explore pathomechanisms, define potential new targets, and screen candidate therapeutic compounds in T2D.

## Figures and Tables

**Figure 1 cells-12-02786-f001:**
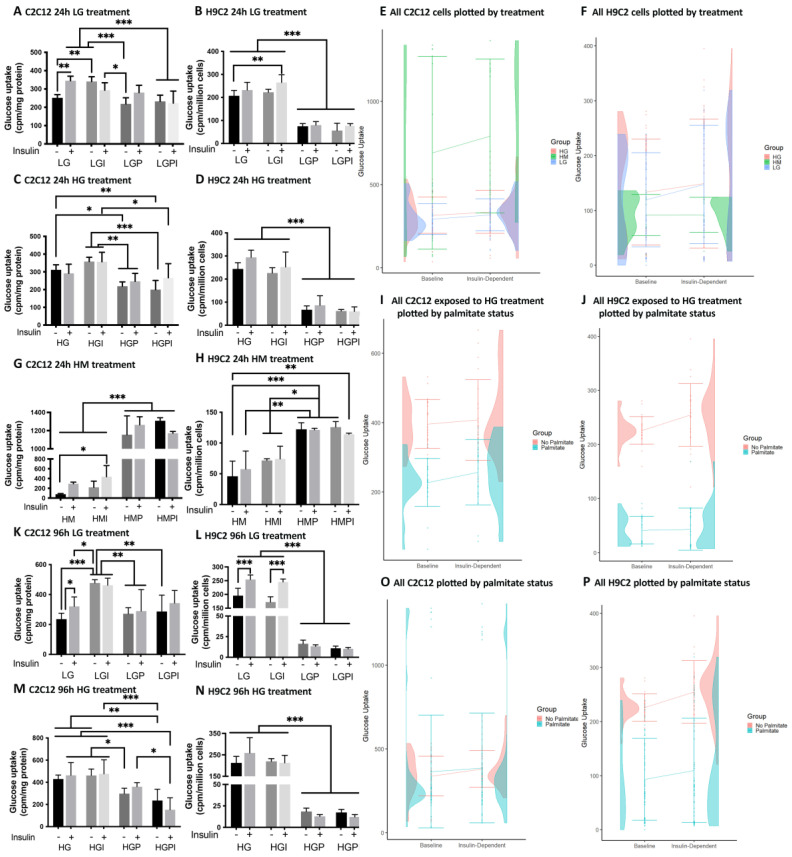
Basal- and insulin-dependent ^3^H-2-deoxy-glucose-uptake of C2C12 and H9C2 myotubes after 24 h and 96 h treatments. The effect upon an acute insulin stimulus (100 nM) was compared between different T2D-mimicking conditions and physiological (LG) and osmotic (HM) controls. IR occurred in all T2D-mimicking conditions. (**E**,**F**,**I**,**J**,**O**,**P**) Half-violin plots for combined 24 h and 96 h baseline and insulin-dependent glucose uptake, with error bars for standard error and lines connecting mean glucose uptake between conditions, by group. (**A**) C2C12 LG control groups after 24 h treatment. (**B**) H9C2 LG control groups after 24 h treatment. (**C**) C2C12 treated with high glucose, palmitate, and insulin for 24 h. (**D**) H9C2 treated with HG, palmitate, and insulin for 24 h. (**E**) All C2C12 cells plotted by treatment. (**F**) All H9C2 cells plotted by treatment. (**G**) C2C12 osmotic control groups treated with HM for 24 h. (**H**) H9C2 osmotic control groups treated with HM for 24 h. (**I**) C2C12 cells receiving high glucose treatment plotted by palmitate status. (**J**) H9C2 cells receiving high glucose treatment plotted by palmitate status. (**K**) C2C12 96 h LG treatment. (**L**) H9C2 96 h LG treatment. (**M**) C2C12 96 h HG treatments. (**N**): H9C2 96 h HG treatments. (**O**) All C2C12 cells plotted by palmitate status. (**P**) All H9C2 cells plotted by palmitate status. Insulin (+): 100 nM acute insulin; LG: low glucose 5 mM; LGI: low glucose + 1 nM insulin; LGP: low glucose + 150 µM palmitate for C2C12 and 75 µM palmitate for H9C2; LGPI: low glucose + 150 µM palmitate for C2C12 and 75 µM palmitate for H9C2 + 1 nM insulin; HG: high glucose 25 mM; HGI: high glucose + 1 nM insulin; HGP: high glucose + 150 µM palmitate for C2C12 and 75 µM palmitate for H9C2; HGPI: high glucose + 150 µM palmitate for C2C12 and 75 µM palmitate for H9C2 + 1 nM insulin; HM: 20 mM mannitol + 5 mM glucose; HMI: 20 mM mannitol + 5 mM glucose + 1 nM insulin; HMP: 20 mM mannitol + 5 mM glucose + 150 µM palmitate for C2C12 and 75 µM palmitate for H9C2; HMPI: 20 mM mannitol + 5 mM glucose + 150 µM palmitate for C2C12 and 75 µM palmitate for H9C2 + 1 nM insulin. Basal glucose uptake and insulin-dependent glucose uptake are presented as cpm per mg protein or per million cells. Values represent the means (SD); n = 6 per treatment group for LG and HG treatments, with n = 3 per group for HM treatments. One-way ANOVA for (**A**–**D**,**G**,**H**,**K**–**N**) *p* < 0.0001. Brackets indicate Tukey’s multiple comparisons tests with * *p* < 0.05, ** *p* < 0.01, *** *p* < 0.001.

**Figure 2 cells-12-02786-f002:**
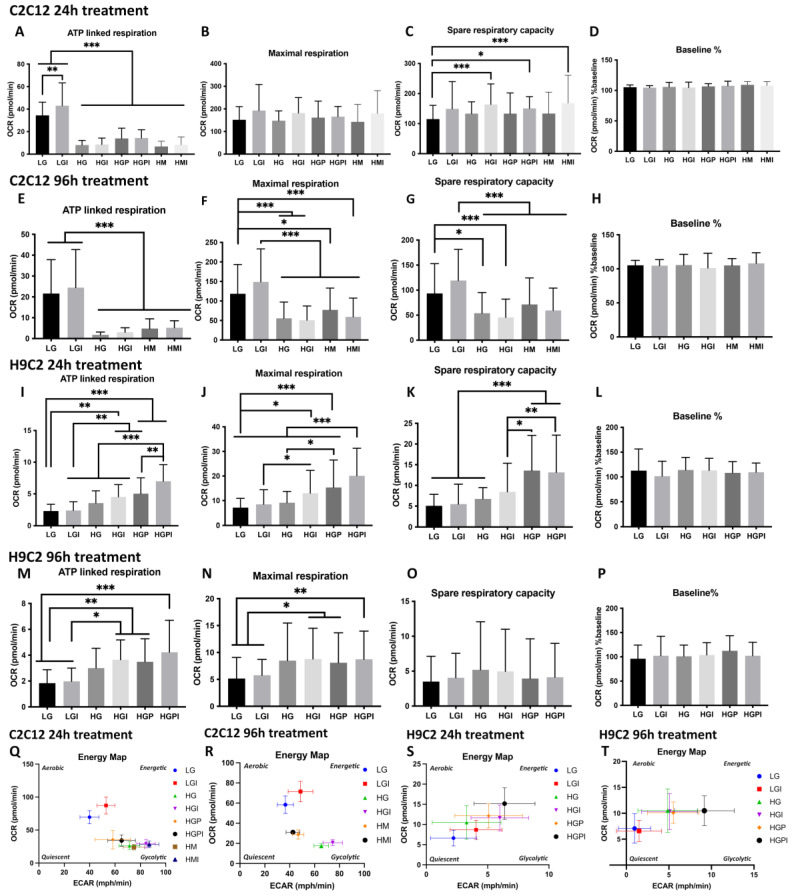
Plate-based oxygen consumption measurements of C2C12 (**A**–**H**,**Q**,**R**) and H9C2 (**I**–**P**,**S**,**T**) myotubes**.** The cells were plated at 1.2 × 10^4^ cells per well in XF96 microplates and differentiated for 5 days. Treatments were carried out for 24 h or 96 h prior to the assay. (**A**) ATP-linked respiration of C2C12 myotubes after 24 h treatments (row/treatment factor (Rf) F (7, 292) = 70.16, *p* < 0.0001; column factor (Cf) F (2, 292) = 9.807, *p* < 0.0001). (**B**) Maximal respiration of C2C12 myotubes after 24 h treatments (Rf F (7, 288) = 2.206, *p* = 0.0338; Cf F (2, 288) = 9.502, *p* = 0.0001). (**C**) Spare respiratory capacity of C2C12 Myotubes after 24 h treatments (Rf F (7, 292) = 3.153, *p* = 0.0031; Cf F (2, 292) = 15.72, *p* < 0.0001). (**D**,**H**,**L**,**P**) Relative oxygen consumption rate percentage (**D**) Rf F (7, 282) = 1.516; Cf F (2, 282) = 5.305, (**H**) Rf F (5, 168) = 0.7035, Cf F (2, 168) = 2.439, (**L**) Rf F (5, 146) = 0.6060, Cf F (2, 146) = 1.126, (**P**) Rf F (5, 131) = 0.8578, Cf F (2, 131) = 0.7791). (**E**) ATP-linked respiration 96 h (Rf F (5, 123) = 21.21, *p* < 0.0001; Cf F (2, 123) = 28.60, *p* < 0.0001). (**F**) Maximal respiration 96 h (Rf F (5, 155) = 19.18, *p* < 0.0001; Cf F (2, 155) = 38.29, *p* < 0.0001). (**G**) Spare respiratory capacity 96 h (Rf F (5, 154) = 12.12, *p* < 0.0001; Cf F (2, 154) = 33.39, *p* < 0.0001). (**I**) ATP-linked respiration of H9C2 myotubes after 24 h treatments (Rf F (5, 139) = 23.58, *p* < 0.0001; incubation factor F (2, 139) = 7.225, *p* = 0.0010). (**J**) Maximal respiration 24 h (Rf F (5, 144) = 20.55, *p* < 0.0001; Cf F (2, 144) = 69.85, *p* < 0.0001). (**K**) Spare respiratory capacity 24 h (Rf F (5, 135) = 15.03, *p* < 0.0001; Cf F (2, 135) = 47.87, *p* < 0.0001). (**M**) ATP-linked respiration 96 h (Rf F (5, 178) = 8.173, *p* < 0.0001; Cf F (2, 178) = 0.2515, *p* = 0.7779). (**N**) Maximal respiration 96 h (Rf F (5, 185) = 5.086, *p* = 0.0002; Cf F (2, 185) = 47.23, *p* < 0.0001). (**O**) Spare respiratory capacity 96 h (Rf F (5, 186) = 0.6408, *p* = 0.6688; Cf F (2, 186) = 26.49, *p* < 0.0001). (**Q**,**R**) OCR plotted versus ECAR of C2C12 after 24 h (**Q**) and 96 h (**R**) treatment. (**S**,**T**) OCR plotted versus ECAR of H9C2 after 24 h (**S**) and 96 h (**T**) treatment. LG: low glucose 5 mM; LGI: low glucose + 1 nM insulin; HG: high glucose 25 mM; HGI: high glucose + 1 nM insulin; HGP: high glucose + 150 µM palmitate for C2C12 and 75 µM palmitate for H9C2; HGPI: high glucose + 150 µM palmitate for C2C12 and 75 µM palmitate for H9C2 + 1 nM insulin; HM: 20 mM mannitol + 5 mM glucose; HMI: 20 mM mannitol + 5 mM glucose + 1 nM insulin. Each graph represents data of minimum of three independent biological replicates presented as means ± standard deviation; n per treatment group = 8–15 (C2C12) or 6–15 (H9C2) wells per assay. Two-way-ANOVA for row factor treatment (Rf) and column factor Seahorse assays (Cf). Significance of secondary pairwise comparisons among treatments by Tukey´s test is indicated by brackets and asterisks *: *p* < 0.05; **: *p* < 0.01; ***: *p* < 0.001 in the panels.

**Figure 3 cells-12-02786-f003:**
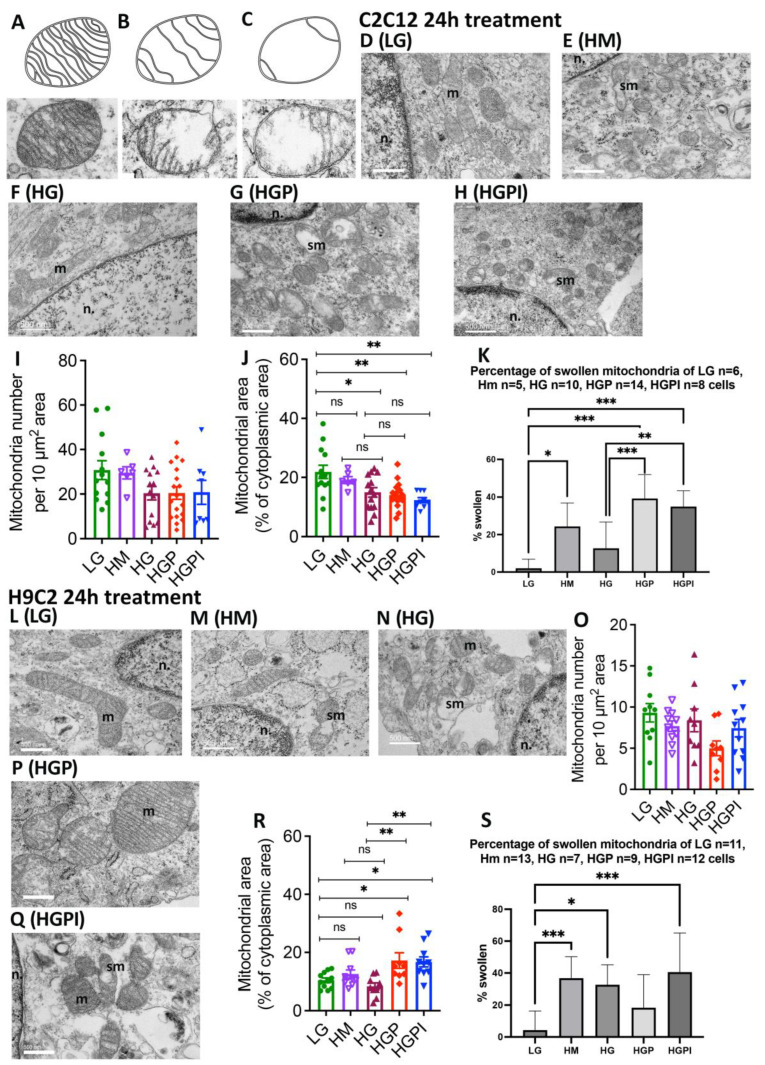
Mitochondrial morphology and number of C2C12 and H9C2 myotubes after 24 h treatment with T2D mimicking conditions. (**A**) Normal muscular mitochondria. (**B**,**C**) Swollen mitochondria. (**D**–**H**) Mitochondria of C2C12 myotubes after different treatments in 15 kX magnification. (**L**–**N**,**P**,**Q**) Mitochondria of H9C2 myotubes; n = nucleus, m = normal mitochondria, sm = swollen/vacuolated mitochondria. (**I**,**O**) Mitochondrial number per 10 µm^2^ area of cytoplasm of C2C12 (**I**) and H9C2 (**O**). (**J**,**R**) Mitochondrial density (area occupied by mitochondria in total area of cytoplasm) in C2C12 (**J**) and H9C2 (**R**). (**K**,**S**) Percentage of swollen mitochondria in C2C12 (**K**) and H9C2 (**S**) myotubes. LG: low glucose 5 mM; HM: 20 mM mannitol + 5 mM glucose; HG: high glucose 24 mM; HGP: high glucose + 150 µM palmitate for C2C12 and 75 µM palmitate for H9C2; HGPI: high glucose + 150 µM palmitate for C2C12 and 75 µM palmitate for H9C2 + 1 nM insulin. One-way ANOVA of (**I**) *p*= 0.1241; (**J**) *p* = 0.0007; (**K**) + (**S**) *p* < 0.0001; (**O**) *p* = 0.0681; (**R**) *p* = 0.0009. Tukey’s tests for multiple comparisons: * *p* < 0.05, ** *p* < 0.01, *** *p* < 0.001.

**Figure 4 cells-12-02786-f004:**
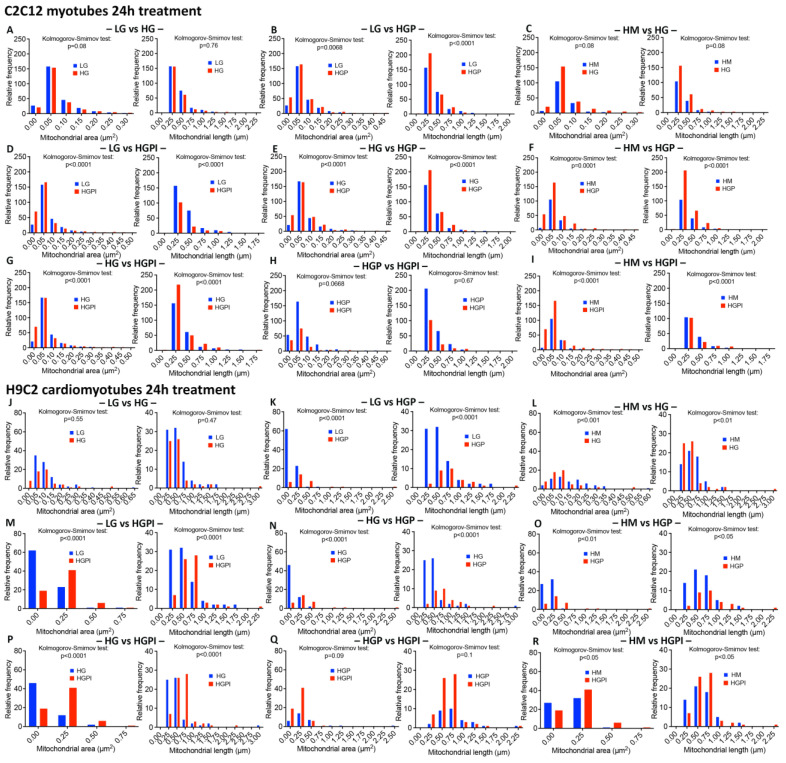
Relative frequency of mitochondria with respect to area and length after 24 h treatment with T2D-mimicking conditions. (**A**–**I**) C2C12 myotubes, (**J**–**R**) H9C2 myotubes. Kolmogorov–Smirnov test of mitochondrial area (µm^2^) and length (µm) of LG vs. HG (**A**,**J**), LG vs. HGP (**B**,**K**), HM vs. HG (**C**,**L**), LG vs. HGPI (**D**,**M**), HG vs. HGP (**E**,**N**), Hm vs. HGP (**F**,**O**), HG vs. HGPI (**G**,**P**), HGP vs. HGPI (**H**,**Q**), and HM vs. HGPI (**I**,**R**). LG: low glucose 5 mM; HM: 20 mM mannitol + 5 mM glucose; HG: high glucose 25 mM; HGP: high glucose + 150 µM palmitate for C2C12 and 75 µM palmitate for H9C2; HGPI: high glucose + 150 µM palmitate for C2C12 and 75 µM palmitate for H9C2 + 1 nM insulin.

**Table 1 cells-12-02786-t001:** Selection of current in vitro models of type 2 diabetes and insulin resistance.

Cell Type and Differentiation	Preincubation	Type of Treatment, Concentration, Duration	Read Outs	Reference
3T3-L1 adipocytes	DMEM 5 mM glucose	Palmitate 0.75 mM 17 h Hypoxia 16 h Dexamethasone 1 µmol/L 24 h High glucose 25 mM 18 h	Inhibition of phosphorylation of insulin receptor and protein kinase B; decrease in insulin dependent glucose uptake Impaired GLUT4 membrane intercalation	[20] [22] [23] [24]
C2C12 myoblasts	DMEM 25 mM glucose	Insulin 60 nM 24 h Palmitate 0.4 mM 24 h	Inhibition of insulin stimulated activation of Akt/protein kinase B; swollen mitochondria	[21]
	DMEM 5 mM glucose	Glucose 15 mM 24 h Palmitate 0.25 mM 24 h	Increased apoptosis, increased ROS production	[25]
C2C12 myotubes	DMEM 5 mM glucose	Palmitate 0.75 mM 17 h	Inhibition of insulin stimulated glycogen synthesis and activation of protein kinase B, diacylglyceride accumulation	[20]
	DMEM not specified	Palmitate 0.6 mM 24 h	Reduced Akt phosphorylation, glucose uptake, and GLUT4 expression	[26]
Huh7 differentiated hepatocellular carcinoma	DMEM 25 mM glucose	Insulin 60 nM 24 h Palmitate 0.4 mM 24 h	Inhibition of insulin stimulated activation of Akt/protein kinase B	[21]
Primary human myotubes	DMEM not specified	Palmitate 0.5 mM 48 h	Decrease in insulin stimulated glucose uptake	[27]
H9C2 myoblasts	DMEM 25 mM glucose DMEM 5 mM glucose	Glucose 33 mM 36 h	Enhanced apoptosis, activation of cardiac hypertrophy proteins	[28]
		Glucose 40 mM 24 h	Increased ROS production + apoptosis	[29]
		Glucose 25 mM + insulin 100 nM 24 h	Decrease in insulin stimulated glucose uptake, inhibition of insulin stimulated activation of Akt	[30]
H9C2 myotubes	DMEM not specified	Palmitate 100 µM 24 h	Decrease in insulin stimulated glucose uptake	[31]

**Table 2 cells-12-02786-t002:** Escalation of palmitate concentrations for long-term (96 h) treatments of C2C12 and H9C2 myotubes. Palmitate concentrations were increased every 24 h.

Treatment Duration	Conc. For C2C12	Conc. For H9C2
24 h	35 µM	5 µM
48 h	70 µM	25 µM
72 h	100 µM	50 µM
96 h	150 µM	75 µM

## Data Availability

All data are available upon request.

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
