# Peer review of "Modeling and Phenotyping Acute and Chronic Type 2 Diabetes Mellitus In Vitro in Rodent Heart and Skeletal Muscle Cells"

_cells, 2023, doi:10.3390/cells12242786_

Round 1

Reviewer 1 Report

Comments and Suggestions for Authors

Kopp et al describe  development of a novel model for simulating T2D in vitro, including hyperglycemia, hyperlipidemia, and variably elevated insulin levels targeting rodent heart and skeletal muscle cells. They mimicked type 2 diabetes by treating C2C12 and H9C2 cells, short- or long-term with insulin and/or palmitate and HG (25 nM; with an osmolarity control) and measuring insulin  resistance (IR), cellular respiration, mitochondrial morphometry and the associated function using glucose uptake, Seahorse flux analyzer and transmission electron microscopy. The authors demonstrate that C2C12 myotubes exposed to T2D -mimicking conditions showed significantly decreased ATP-linked respiration and mitochondrial dysfunction. In contrast, H9C2 myotubes showed elevated ATP-linked and increased cytoplasmic area occupied by mitochondria, indicating better adaptation of these cells to stress and compensatory lipid oxidation. Interestingly, both cell types showed elevated fractions of swollen/vacuolated mitochondria after T2D -mimicking treatments.  The study is well done and the authors convincingly demonstrated the effect of T2D conditions in cell culture in vitro as well as contrasting effects on different cell types.  This is an excellent and timely study and may help in understanding molecular mechanisms and novel therapeutics for T2D. 

Minor concern:

Use space between number and concentration (25 mM instead of 25mM).

Author Response

Dear Reviewer,

Thank you very much for taking the time to review our manuscript. Thank you for suggesting to use space between number and concentration (25 mM instead of 25mM), which we updated in the re-submitted file.

Best regards,

Elena Kopp

Reviewer 2 Report

Comments and Suggestions for Authors

The present study evaluates in vitro models of insulin resistance in skeletal and cardiac muscle cells induced by a combination of hyperglycemia, hyperinsulinemia, and hyperlipidemia. The authors use established methods of glucose uptake assays to measure fuel uptake at basal and insulin-induced states of these cells after 24 or 96 hours of treatment, extracellular flux analysis to assess mitochondrial function and electron microscopy to evaluate mitochondrial morphology in the model systems. The study of Kopp et al. attempts to address the lack of well-characterized in vitro models of type-2 diabetes that would better reflect the disease phenotype observed in patients with T2DM. The overall study design, the data presentation and the conclusions drawn, however, would need significant improvement before publication. Please, see the major concerns listed in the comments below:

1.)   It is not clear why the authors chose the palmitate concentrations used for the treatment of the skeletal and cardiac muscle cells. Demonstration of the effect of the treatment on cell viability and insulin signaling pathway activation as a validation of the model is needed, especially as 50uM PA treatment of H9C2 cells for 24h already decreases cell viability (Chen et al. J. of Nutritional Biochemistry 2016). Is there published data available on this specific treatment protocol?

2.)   Why omit the palmitate-treated low glucose samples (LGP and LGPI)  as well as the mannitol/palmitate (HMP and HMPI) controls from the mitochondrial function analysis for the C2C12 24h treatment? As the paper progresses more and more of the appropriate controls are left out from the analyses, undermining the validity of the results and the conclusions drawn. This is a serious experimental design flaw.

3.)   Similarly, why were the high mannitol controls left out from mitochondrial structural analysis? Even if, as the authors state, there were no changes detected in mitochondrial function in these samples, that does not necessarily mean that mitochondrial number or density was not affected in these samples.

4.)   Mitochondrial morphology was not analyzed for cells treated for 96 hours. Why the exclusion?

5.)   When analyzing mitochondrial area and length, data presentation should be reconsidered to allow for comparison between the treatment and all of the appropriate controls, not only log glucose (LG) conditions, as that comparison is not appropriate when multiple treatments are introduced – ie. high glucose with palmitate and insulin.

Minor comments:

Line 53: “T2D results in extremely robust insulin insensitivity.” – Please, review, as in its current form, this statement is debatable.

Line 95: Were the cells starved for 4 or 6 hours? It is an important detail to ensure reproducibility.

Line 111: were the cells serum starved for 3 or 4 hour before glucose uptake assays?

Lines 131-150 is a verbatim repetition of lines 111-131.

Line 211 – reference # 12 seems to be inappropriate here.

Line 292 – mentioning glycolytic shift – presented as ECAR on Fig2, but ECAR measurements were not detailed in the methods.

Line 311 – Should be: “3.2.2 H9C2”, not “3.2.1 C2C12”

Line 437 – “C2C12 myotubes …showed…more ROS production…” ROS was not measured or at least not presented in the results.

Line 438 “…however, ATP-linked respiration was not altered…” – Figure 2A shows a significant difference in ATP-linked respiration of LG vs HG-treated C2C12 cells.

Line 454 – “FA uptake into the heart is not hormonally controlled” – consider review – insulin participates in long-chain fatty acid uptake in heart muscle.

Line 468 – for clarity, please, consider introducing a definition of mitochondrial density in methods to improve clarity.

Line 491 – “ For future experiments a mixture (i.e. palmitate and oleate) could be used.” – For consideration: palmitate and oleate differently modulate insulin sensitivity in human muscle.

Figures – graphs in figures do not match up with the figure legends – and sometimes the text. Ie. Figure 1, (Line 254) C2C12 data is shown in Fig. 1 (A, C, E, G, I, K, M, O) not in (A-G, I) as indicated in Figure 1 legend. This is just one of the MANY mismatches.

Figure 2 : No § symbols found on graphs – not clear why they are listed in the figure legend.

Figure 3 – larger electron micrographs would be appreciated by all readers – it is challenging to see any changes in the mitochondria on the current pictures.

The size of scale bars is barely visible in the image. Consider including in the figure legend.

Figure 4 – using the same length/area brackets for all graphs would significantly aid comparison between groups.

Reviewer 3 Report

Comments and Suggestions for Authors

The study by Kopp et al. report a novel model for studying type 2 diabetes mellitus in vitro. On the basis of the results obtained, the authors conclude that the model should allow studying disease mechanisms, potential new targets, and 35 screen candidate therapeutic compounds. The manuscript is well-written and well-organized. The results obtained are interesting although further investigations are needed. However, some points should be addressed.

- Title: Diabetes in a dish is a kind of expression more appropriate for review. I suggest to delete this expression.

- The Authors must provide all the statistical values of the ANOVA. To be more precise, the Authors must provide the F values of the ANOVA throughout the manuscript.

- The introduction is too short. More studies supporting the rationale for conducting this study must be added and discussed. For instance, studies (e.g. PMID: 34429169; PMID: 34206340 ) that have tried to model type 2 diabetes mellitus should be reported and discussed.

-There are typos throughout the manuscript that should be corrected.  

Comments on the Quality of English Language

Minor editing.

Round 2

Reviewer 2 Report

Comments and Suggestions for Authors

Thank you for your response. I will include previous comments and answers and highlight the new ones.

Major Comments:

Comments 1: It is not clear why the authors chose the palmitate concentrations used for the treatment of the skeletal and cardiac muscle cells. Demonstration of the effect of the treatment on cell viability and insulin signaling pathway activation as a validation of the model is needed, especially as 50uM PA treatment of H9C2 cells for 24h already decreases cell viability (Chen et al. J. of Nutritional Biochemistry 2016). Is there published data available on this specific treatment protocol?

Response 1: Thank you for pointing this out. For H9C2 we chose a final concentration (75 µM) on the lower threshold of previously used concentrations (typically 100-400 µM) that can induce lipotoxicity (Hsu 2016 https://doi.org/10.1007/s00394-015-1034-7). Zou et al showed that palmitate decreased cell viability in a dose-dependent manner but 100 µM did not induce significant cell viability loss (Zou 2017 doi:10.3892/mmr.2017.7404). Palmitate (100 µM) induces insulin resistance (Chang 2016 doi:10.1016/j.bbalip.2015.12.017). At lower palmitate concentrations (max 50µM) we did not observe significantly impaired glucose uptake in test assays (lower n), which is why we increased to a final concentrations of 75µM. For C2C12 we were aiming to show that much lower concentrations of palmitate than previously used (see Table 1) can also induce IR and are closer to the physiological range in rodents (100-400 µM, Nawrocki A 2004). In the revised manuscript we added explanations in the introduction (lines 111-116).

Given the conflicting data of published literature on what concentration of palmitate decreases cell viability (50 or 100 µM) it is this reviewer’s opinion, that inclusion of data demonstrating cell viability upon the palmitate treatments employed is crucial for the transparency of the work presented and validity of the model.

Also, the authors state on Line 114: “We therefore worked with a maximal concentration of 150 μM for C2C12 and 75 μM for 114 H9C2, which still leads to IR but is less toxic and close to the physiological range (300-410 μM in humans [40] and 100-400 μM in rats [41]).

How about in mice? The C2C12 cells are of murine origin. And if these are plasma concentrations how do they compare to interstitial fatty acid concentrations in normal and obese animals? These palmitate concentrations are closer to the lower end of the physiological concentrations per the authors, but still result in IR in the cell culture setting. How can the authors explain that? Overall, it is this reviewer’s opinion that this argument does not sufficiently justify the palmitate concentrations chosen.

Comments 2: Why omit the palmitate-treated low glucose samples (LGP and LGPI) as well as the mannitol/palmitate (HMP and HMPI) controls from the mitochondrial function analysis for the C2C12 24h treatment? As the paper progresses more and more of the appropriate controls are left out from the analyses, undermining the validity of the results and the conclusions drawn. This is a serious experimental design flaw.

Response 2: The effect was less robust for the low glucose groups with varied treatments at lower n values of data not shown. We therefore completed studies largely with the high glucose groups and only included the basic low glucose groups as a comparison.

If the n is sufficiently high to ensure adequate statistical power for these experiments, then why not include them for a more thorough analysis? If the n is not high enough, any conclusions on the effect’s robustness would be questionable. Omission of these samples is still an experimental design flaw in this reviewer’s opinion.

Comments 4: Mitochondrial morphology was not analyzed for cells treated for 96 hours. Why the exclusion?

Response 4: We analyzed a lower n (not enough for publication) of cells treated for 96 hours for mitochondrial morphology and did not observe major differences to 24 h treatments.

Since we already observed the changes at 24 hours, we did not further pursue the studies of morphology at 96 hours.

Like before, if the n is sufficiently high to ensure adequate statistical power for these experiments, then why not include them for a more thorough analysis? If the n is not high enough for an acceptable statistical power, any conclusions on any differences would be questionable. Inclusion of these samples would be necessary in this reviewer’s opinion.

Comments 5: When analyzing mitochondrial area and length, data presentation should be reconsidered to allow for comparison between the treatment and all of the appropriate controls, not only low glucose (LG) conditions, as that comparison is not appropriate when multiple treatments are introduced – ie. high glucose with palmitate and insulin

Response 5: Our response provided to Comment 2 applies also for the analysis of mitochondrial area and length. However we now extended the analysis with a higher n of HM-treatment electron micrographs and added the comparisons to figure 4.

Inclusion of novel data with HM-treated samples does not address the fact that the comparisons performed often don’t include the right controls. For example, comparing low glucose treatment to a treatment with high glucose+palmitate+insulin cannot tease out the differential effects of the individual treatment. Response to Comment 2 does not really address this concern either.

Minor Comments:

Line 53: “T2D results in extremely robust insulin insensitivity.” – Please, review, as in its current form, this statement is debatable. Agree, the revised manuscript you can find a reviewed statement in line 52.

Line 51:T2D results in severe insulin insensitivity.” While the statement was rephrased, this is still debatable: insulin resistance is a prelude to the development of T2DM, not the other way around. T2D is characterized by insulin resistance but does not lead to it.

Line 454 – “FA uptake into the heart is not hormonally controlled” – consider review – insulin participates in long-chain fatty acid uptake in heart muscle. Agree, the revised manuscript you can find a reviewed statement (line 523).

Line 523 now reads:Unlike most glucose uptake is insulin dependent, FA uptake into the heart is not dependent on hormones and mainly driven by the availability in the blood stream [67].”

Fatty acid uptake into the heart can still be controlled by insulin signaling, so this statement in its current form is still not correct.

Additional comments on the revised manuscript:

Line 76: “ …an excess of plasma FFA can cause ischemic damage and high oxidative stress [14].”

Increased fatty acid uptake/use will predispose the heart to ischemic damage, but in itself will not cause it. Revision of the statement is suggested.

Line 91: “… once a cell becomes overwhelmed by the excess supply of FFAs, toxic lipids can be generated, such as diacylglyceride (DAG)…”

Consider revising this statement, as DAG in itself is not toxic, indeed it is an important signaling intermediate. Its accumulation leads to toxicity.

Line 96: “ … more likely reflect acute or uncontrolled T1D.” – T1D is a chronic disease, please, consider rephrasing.
